Myofascial release versus Mulligan sustained natural apophyseal glides’ immediate and short-term effects on pain, function, and mobility in non-specific low back pain

Bhat P Vignesh 1
Patel Vivek Dineshbhai vivek.patel@manipal.edu 1
Eapen Charu 1
Shenoy Manisha 2
Milanese Steve 3
1 Department of Physiotherapy, Kasturba Medical College, Mangalore, Manipal Academy of Higher Education , Karnataka , India
2 Department of Physical Therapy, Hamad Medical Corporation , Doha , Qatar
3 International Centre for Allied Health Evidence, University of South Australia , Adelaide , South Australia , Australia
Abdala Virginia
Electronic publication date: 2021 Mar 15
Publication date: 2021
Volume: 9
Electronic Location ID: e10706
Received 2020 Aug 12; Accepted 2020 Dec 14
Copyright: ©2021 Bhat P et al.
Copyright year: 2021
Copyright holder: Bhat P et al.
License: This is an open access article distributed under the terms of the Creative Commons Attribution License, which permits unrestricted use, distribution, reproduction and adaptation in any medium and for any purpose provided that it is properly attributed. For attribution, the original author(s), title, publication source (PeerJ) and either DOI or URL of the article must be cited.
License URL: https://creativecommons.org/licenses/by/4.0/

Keywords: Mulligan SNAGs, Myofascial release, Non-specific low back pain, Strengthening exercises

Funding: The authors received no funding for this work.

==============================
Background

Myofascial release (MFR) and Mulligan Sustained Natural Apophyseal Glides (SNAGs) are manual therapy techniques routinely practiced in the management of non-specific low back pain (NSLBP). As a solo intervention or along with other therapies, both methods have reported positive results for individuals with NSLBP. However, which technique improves NSLBP-related pain, restricted range of motion (ROM) and disability, warrants further research.

Objective

To study the comparative effects of MFR and SNAGs on pain, disability, functional ability, and lumbar ROM in NSLBP.

Method

A parallel-group study was conducted at tertiary care hospitals. Sixty-five Sub-acute or chronic NSLBP patients were allocated to receive strengthening exercises along with either MFR (n = 33) or SNAGs (n = 32) for six treatment sessions over one week. An independent assessor evaluated outcome measures such as the Visual Analog Scale (VAS), Patient-Specific Function Scale (PSFS), and ROM at baseline, immediate (after 1st treatment), and short-term (post-sixth day of the intervention). The Modified Oswestry disability index (MODI) was assessed at baseline and short-term.

Results

Within-group analysis found clinically and statistically significant (p < 0.05) changes for VAS and PSFS at immediate and short-term for both the groups. The lumbar extension also showed improvement immediately and in the short-term. Improvement in Lumbar flexion was seen only in the SNAGs group over the short-term. A statistically significant improvement was seen for MODI in both the groups but was not clinically significant in the MFR group. The analysis observed no statistically significant difference (p < 0.05) between the groups at both the immediate and short-term.

Conclusions

Pain and restricted function associated with NSLBP can be improved using SNAGs or MFR, along with strengthening exercises. For limited lumbar flexion ROM, Mulligan SNAGs have a better outcome than MFR over the short-term. Hence, both manual therapy techniques can be incorporated along with exercises for immediate and short-term management of sub-acute to chronic NSLBP.

Clinical Trial Registration. CTRI/2018/12/016787 (http://ctri.nic.in/Clinicaltrials/).

Introduction

Low back pain (LBP) is a debilitating health condition, ranked first in musculoskeletal disease burden worldwide (GBD 2017 Disease and Injury Incidence and Prevalence Collaborators, 2018; Hoy et al., 2014). It is reported to have an 18.3% mean point prevalence and 30.8% one-month prevalence (Maher, Underwood & Buchbinder, 2017). According to the Global Burden of Disease study, LBP emerged as a primary cause for years lived with disability (YLD) for all age groups in both sexes (GBD 2017 Disease and Injury Incidence and Prevalence Collaborators, 2018). From 1990 to 2007, YLDs due to LBP increased by 30%, with a further increase of 17% in the last decade (GBD 2017 Disease and Injury Incidence and Prevalence Collaborators, 2018).

Approximately 10% of LBP cases have an identifiable pathology, while the remaining 90% are non-specific LBP (NSLBP), reflecting LBP of unknown underlying pathology, characterized by pain, muscle tension, and stiffness between 12th rib and inferior gluteal fold (Maher, Underwood & Buchbinder, 2017). Based on duration, LBP can be categorized as acute (less than six weeks), sub-acute (six to twelve weeks), and chronic (more than twelve weeks) (Krismer, 2007).

One proposed mechanism underpinning NSLBP involves changes in lumbosacral proprioception and core muscle recruitment patterns due to atrophy of the lumbar stabilizers (Goubert et al., 2016) and gluteus maximus (Jeong, Kim & Hwang-Bo, 2015) along with other hip muscles weakness (De Sousa et al., 2019). The gradual decrease in motor control leads to uncontrolled and abnormal tissue loading on the myofascial complex, (Ajimsha, Daniel & Chithra, 2014; Tozzi, Bongiorno & Vitturini, 2011) stressing the lumbar spine leading to pain (Goubert et al., 2016).

The primary line of management for NSLBP includes analgesics and physical therapy interventions (Maher, Underwood & Buchbinder, 2017; Van Middelkoop et al., 2011). The routine physical therapy interventions are transcutaneous electrical nerve stimulation, low-level LASER therapy, manual therapy, (Pourahmadi et al., 2018) back schools, exercise, and timely review (Van Middelkoop et al., 2011; Pourahmadi et al., 2018). Despite the range of interventions available, NSLBP leads to chronic loss of health by limiting activity participation and loss of function, potentially resulting in prolonged work disability (Krismer, 2007). Manual therapies such as Mulligan mobilization, (Hussien et al., 2017; Hidalgo et al., 2015; Waqqar, Shakil-Ur-Rehman & Ahmad, 2016; Khan, Torairi & Shamsi, 2018; Elrazik et al., 2016; Tul Ain et al., 2019; Muhanna, 2018) McKenzie exercises, (Waqqar, Shakil-Ur-Rehman & Ahmad, 2016) Maitland mobilization, (Khan, Torairi & Shamsi, 2018; Elrazik et al., 2016), and Myofascial release therapy (MFR) (Ajimsha, Daniel & Chithra, 2014; Tozzi, Bongiorno & Vitturini, 2011; Saratchandran, 2013; Arguisuelas et al., 2017; Arguisuelas et al., 2019; Balasubramaniam, Ghandi & Sambandamoorthy, 2013) are used routinely in clinical practice for the management of NSLBP.

The Mulligan concept is based on the theory that minor positional faults of articulating joints’ surfaces following injury or strain result in a painful and restricted, range of motion (ROM) (Pourahmadi et al., 2018; Hussien et al., 2017). Mulligan Sustained Natural Apophyseal Glides (SNAGs) technique adds a passive accessory glide, parallel to the joint plane using a vertebral spinous process or transverse process, during which the patient performs the previously painful or restricted active movement (Hussien et al., 2017; Hidalgo et al., 2015; Waqqar, Shakil-Ur-Rehman & Ahmad, 2016; Khan, Torairi & Shamsi, 2018; Elrazik et al., 2016; Tul Ain et al., 2019; Muhanna, 2018; Moutzouri et al., 2008). Stimulation of mechanical receptors by spinal mobilization activates large-diameter nerve fibers leading to activation of the pain gate mechanism (Rezkallah & Abdullah, 2018). At the central level, descending pain pathways may be facilitated via the midbrain’s periaqueductal grey matter (Rezkallah & Abdullah, 2018). These descending neurons may release the primary mediators’ opioids, nor-adrenaline, and serotonin, modulating pain, reducing the muscle spasm, and improving restricted lumbar movements (Elrazik et al., 2016; Rezkallah & Abdullah, 2018).

A systematic review indicated moderate level evidence of Mulligan technique for short-term effect on LBP associated pain and disability (Pourahmadi et al., 2018). Studies in this review delivered SNAGs in addition to conventional therapy, including stretching and back extensor strengthening (Hussien et al., 2017; Khan, Torairi & Shamsi, 2018; Elrazik et al., 2016), and thoracic postural exercises (Khan, Torairi & Shamsi, 2018; Tul Ain et al., 2019). Immediate and short term benefits of SNAGs as a standalone treatment is reported in patients with NSLBP (Hidalgo et al., 2015) as well as healthy individuals (Moutzouri et al., 2008) compared to sham SNAGs. Mulligan SNAGs found superior to McKenzie extension exercises to improve lumbar ROM but not for pain and disability in patients with NSLBP (Waqqar, Shakil-Ur-Rehman & Ahmad, 2016).

Myofascial release is a manual technique that utilizes a superintend force in a predetermined direction to stretch or optimize the myofascial complex’s length and gliding properties (Ajimsha, Daniel & Chithra, 2014; Tozzi, Bongiorno & Vitturini, 2011). MFR improves myofascial restriction by breaking intermolecular cross-links and redistributing internal fluids (Ajimsha, Daniel & Chithra, 2014; Tozzi, Bongiorno & Vitturini, 2011; Saratchandran, 2013; Arguisuelas et al., 2017; Arguisuelas et al., 2019; Balasubramaniam, Ghandi & Sambandamoorthy, 2013; Rezkallah & Abdullah, 2018). The prolonged-release in MFR superimposes stretch over joint and muscle mechanoreceptors (Balasubramaniam, Ghandi & Sambandamoorthy, 2013). These mechanoreceptors activate the sympathetic system by somatic efferent and periaqueductal grey matter modulating the descending pain pathway (Saratchandran, 2013; Balasubramaniam, Ghandi & Sambandamoorthy, 2013).

Two systematic reviews suggested emerging evidence of MFR for chronic LBP (Ajimsha, Al-Mudahka & Al-Madzhar, 2015); however, the observed effect was not clinically significant (Laimi et al., 2018). In these reviews, MFR was given as an adjunct to specific back exercises (Ajimsha, Daniel & Chithra, 2014) and occupational therapy (Saratchandran, 2013) or compared to sham intervention (Tozzi, Bongiorno & Vitturini, 2011) in NSLBP patients. Improvement was observed for pain, fascial mobility, and functional abilities following MFR intervention among non-specific neck and back pain patients (Tozzi, Bongiorno & Vitturini, 2011). MFR as a standalone treatment improved pain, performances of daily activities, and fear of pain (Arguisuelas et al., 2017) and lumbar ROM in patients with NSLBP over the short-term (Arguisuelas et al., 2019).

Both MFR and SNAGs have shown beneficial effects in managing NSLBP. However, a dearth of evidence about the comparative effect of MFR and SNAGs as an adjunct to strengthening exercises in NSLBP warrants further research. Hence, this study sought to compare the (immediate and short term) effects of MFR and SNAGs as adjunct treatments in patients with NSLBP.

Materials & Methods

The parallel-group study was carried out at tertiary care hospitals from November 2018 to March 2020. Institutional Ethics Committee, Kasturba Medical College, Mangalore granted ethical approval (IEC KMC MLR 11-18/429) to carry out the study. The study design was registered under the clinical trial registry of India, https://ctri.nic.in with identifier CTRI/2018/12/016787. Written and oral instructions about the study procedures, interventions, and possible benefits and risks were given to the patients. Written informed consent was taken from all the patients. Patients were assigned to either intervention group in an alternate sequence at a 1:1 ratio. The therapist, who delivered the intervention, did a non-concealed allocation of the patients. As it is an inherent issue to manual therapy trials, the therapist could not be blinded to the patient’s group allocation. However, patients were blinded to the other intervention group.

Patients

Patients referred by orthopedic surgeons for physiotherapy were recruited. Patients with localized back pain with restricted/painful lumbar spine movements were screened for eligibility. The inclusion criteria were sub-acute to chronic NSLBP, either gender, 18–60 years old, and a minimum baseline Visual Analog Scale (VAS) score of four (Amundsen et al., 2018). Patients with contraindications to manual therapy interventions were excluded if they presented with lumbar radiculopathy, spinal pathology (fracture or tumors) or history of any spinal surgery, lumbar canal stenosis, osteoporosis, pregnancy-related back pain, and spinal deformities like scoliosis or kyphosis (Hussien et al., 2017; Rezkallah & Abdullah, 2018; Amundsen et al., 2018). Sixty-five patients with sub-acute to chronic NSLBP were included in the study after screening for eligibility criteria. After screening for eligibility, patients were examined for active lumbar ROM to identify involved painful/restricted segment, which was confirmed using passive accessory intervertebral movement examination in a prone position. Patients’ flow is highlighted in the consort flow diagram (Fig. 1).

Figure 1 CONSORT flow diagram.

Outcome measures

An independent blinded assessor collected all the outcome data from the patients at baseline, immediately post first treatment session except Modified Oswestry disability index questionnaire, and after the sixth day of intervention (short term).

Pain levels were assessed with the VAS. It is a 100 mm horizontal scale with ’no pain’ and ’worst possible pain’ labels at the line’s extremes. The VAS has demonstrated good test-retest reliability, which is higher among literate (r = 0.94, p < 0.001) than illiterate (r = 0.71, p < 0.001) subjects (Hawker et al., 2011).

Patient-Specific Function scale (PSFS) was used to assess functional ability. The patient was asked to write down three activities that were the most restricted or challenging to perform. All the activities were scored on a scale of zero to ten, where ’zero’ is unable to perform/challenging to do, and ’ten’ can do as before. Previous research on the PSFS has reported moderate to good reliability with an intraclass correlation coefficient (ICC) of 0.713, and a minimal detectable change (MDC) of three, and minimal important difference (MID) of 1.2 (Hefford, 2012).

Disability assessment was measured using the MODI questionnaire, which has ten sections and provides information on LBP’s effect on the patient’s ability to manage everyday life. A total score was converted to percentage points. Fritz and Irrgang (2001) reported a high test-retest reliability of the MODI in 67 LBP patients with an ICC of 0.90 and a minimum clinically important difference (MCID) of six percentage points (Fritz & Irrgang, 2001).

Range of motion was assessed using a bubble inclinometer, as described by Norkin & White (2009). A study on the within and between-day bubble inclinometer reliability in determining standing lumbar spine ROM (Flexion, extension, and lateral flexion) in healthy individuals and chronic NSLBP patients found ICCs ranged from 0.908 to 0.982 (Sadeghi et al., 2015). Inferiorly S2 and superiorly T12 spinous process was used for double inclinometer measurement technique. The patient was instructed to perform active lumbar movements without bending knees.

Intervention

A total of six intervention sessions in a week were delivered to all participants in both groups.

Procedure for SNAGs (Mulligan, 2010)

The flexion or extension glide application was decided based on the movement examination for restricted lumbar ROM and pain response. The SNAGs were applied in the sitting position with the patient’s pelvic stabilized with a Mulligan belt at the level of the anterior superior iliac spine (ASIS). The therapist’s hand’s ulnar aspect was used over the spinous process of the superior vertebra of the involved segment for flexion glide and the inferior vertebra’s spinous process for extension glide. A passive accessory glide was administered and maintained until the patient completed a full movement arc, which was restricted or painful earlier. The glides were given for six repetitions for three sets every session. (Fig. 2). The glide was administered over the spinous process, where the force’s amplitude was upheld within the patient’s comfort, as Mulligan has previously described that SNAGs should not provoke any pain.

Figure 2 Mulligan SNAGs technique.

(A)- Starting Neutral Position (B) - Neutral to Extension SNAGs.

Procedure for MFR (Saratchandran, 2013; Ajimsha, Al-Mudahka & Al-Madzhar, 2015)

The patient was positioned comfortably in prone lying. Direct MFR was administered to the lower back muscles with the therapist’s knuckles, and the stretch held into the end range for up to 120 s or until the therapist felt giving away the taut tissues before being released. MFR was delivered for ten repetitions, with a total of 20 min of intervention. (Fig. 3).

Figure 3 Myofascial release technique.

Strengthening exercises (Jeong, Kim & Hwang-Bo, 2015; De Sousa et al., 2019; Sadler et al., 2019)

Strengthening exercises were prescribed for all the patients, according to the referring orthopedic surgeon’s direction. Both groups received strengthening exercises with two sets of ten repetitions without any additional resistance. Abdominal draw-in manoeuvre to activate transverse abdominis was performed in crook lying. Cat and camel exercises were carried out for lumbar multifidus training in a quadruped position. Strengthening of gluteal muscles (hip abductors and extensors) was performed in a side-lying and prone lying positions with straight leg raise exercises without any additional resistance. Patients were also briefed about ergonomic advice on posture and lifting techniques to incorporate during routine activities.

Power calculation

The sample size was calculated using the G*Power analysis software (version 3.0.10). The effect size for VAS was estimated, d = 20 mm, and standard deviation (σ) = 26.5 mm from a previous study (Waqqar, Shakil-Ur-Rehman & Ahmad, 2016). With a power of 85% and α level of 0.05 total sample size estimated to be seventy (35 in each group) considering a 10% dropout rate.

Statistical analysis

An independent statistician analyzed data using SPSS Version 25.0 (SPSS Inc, Chicago, IL, USA). Data were assessed for normality using skewness and kurtosis values and observation of Q–Q plots, which indicated that non-parametric tests were required. The demographic characteristics of the patients were summarized with median and interquartile ranges. Data for the lost to follow-up patients were analyzed using the intention to treat analysis by carrying forward the value of outcome measures assessed at the last follow-up. p values less than 0.05 were considered statistically significant. The Friedman rank-sum test was used to determine the within-group differences from baseline to post-treatment sessions for VAS, PSFS, and lumbar ROM. Post-hoc analysis using the Wilcoxon signed-rank test with Bonferroni correction was performed for time*group differences. Mann–Whitney U test was used to explore the two groups’ differences from baseline to immediate, and short-term.

Results

Throughout the trial phases, patients’ flow is highlighted in the consort flowchart (Fig. 1). One hundred and sixty-seven patients were screened for eligibility, of which 102 patients were excluded based on exclusion criteria. Sixty-five patients could be recruited within the study period and allocated to either MFR (n = 33) and SNAGs (n = 32) groups. Eight patients dropped out before the sixth session, either because they dramatically improved and discharged or migrated. Intention to treat analysis was considered to accommodate dropouts. The demographic data of all the participants are shown in Table 1. Baseline characteristics of all the outcome measures were homogenous and statistically insignificant between the groups. (Table 2) The within-group analysis identified statistically significant differences for VAS, PSFS, and extension ROM for both the groups and only for flexion ROM in the SNAGs group. (Table 3) Modified Oswestry disability index also showed statistically significant (p < 0.05) improvement from sixteen and fourteen points at baseline in MFR and SNAGs groups respectively to twelve and eight points over the short-term.

Table 1 Demographic details of the participants.

Variables	MFR (n = 33)	SNAGs (n = 32)	
Age (years, Mean ±  SD)	25 ±   7.11	24.34 ±  5.37	
Gender	Male	15	5	
Female	18	27	
Duration of LBP(weeks)	Sub-acute (6–12 weeks)	2	5	
Chronic (>12 weeks)	31	27	
Notes.

n Total number of participants

SD Standard Deviation

LBP Low back pain

Table 2 Baseline characteristics of VAS, MODI, PSFS and ROM for MFR and SNAGs groups.

Variables	MEDIAN (IQR)	p-value	
	MFR	SNAGs		
VAS (mm)	6.2 (5.2–7.2)	6.1(4.5–4.7)	0.11	
MODI (%)	16(12–25)	14(1.5–25)	0.545	
PSFS	4.33(3.83–5.33)	4.33(3.33–5.33)	0.232	
Flexion (degrees)	50(45–57.5)	50 (44.25–57.75)	0.712	
Extension(degrees)	18 (10–20)	16.5 (10.25–25)	0.889	
Left lateral flexion(degrees)	20 (15–25)	20 (15–28.5)	0.595	
Right lateral flexion(degrees)	20 (13–25)	20 (15–25)	0.633	
Notes.

* p < 0.05 statistical significant.

IQR Inter quartile Range

MFR Myofascial release

SNAGs Sustained Natural Apophyseal Glides

VAS Visual Analog Scale

mm millimetre

MODI Modified Oswestry Disability Index

PSFS Patient Specific Functional Scale

Table 3 Within-group analysis of VAS, PSFS, and ROM for MFR and SNAGs groups at Baseline, immediate and short-term.

Variable	Group	MEDIAN (IQR)	Friedman Chi-square	p-value	
		Baseline	Immediate	Short-term			
VAS (mm)	MFR	6.2 (5.2–7.2)	4.1(1.9–5.2)	2(1.0–3.45)	49.465	<0.001*	
SNAGs	6.1(4.5–4.7)	3.05(2.05–4.37)	2(0.85–3.65)	53.382	<0.001*	
PSFS	MFR	4.33(3.83–5.33)	5.66(4.49–6.16)	7(5.83–7.58	55.197	<0.001*	
SNAGs	4.33(3.33–5.33)	5.33(5–6.66)	6.83(6.12–7.62)	49.589	<0.001*	
Flexion(degrees)	MFR	50(45–57.5)	50 (45–59)	52 (45.5–60)	3.227	0.199	
SNAGs	50 (44.25–57.75)	52.5 (45–59.5)	55.5 (50–60)	6.660	0.035*	
Extension(degrees)	MFR	18 (10–20)	20 (15–30)	25 (19–30)	30.624	<0.001*	
SNAGs	16.5 (10.25–25)	21.5 (18.25–29.5)	25 (20–31.5)	22.505	<0.001*	
Left lateral flexion
(degrees)	MFR	20 (15–25)	22 (15–30)	20 (20–25)	0.890	0.640	
SNAGs	20 (15–28.5)	20 (15–25)	24.5 (15–28)	2.347	0.309	
Right lateral flexion(degrees)	MFR	20 (13–25)	20 (17.5–26)	20 (17.5–26)	5.957	0.050	
SNAGs	20 (15–25)	20 (15–27.75)	20 (15.75–28)	3.588	0.166	
Notes.

* p < 0.05 statistical significant.

IQR Inter quartile Range

VAS Visual Analog Scale

mm millimetre

MFR Myofascial release

SNAGs Sustained Natural Apophyseal Glides

PSFS Patient Specific Functional Scale

Time*group: For both the groups, VAS and PSFS showed immediate and short-term improvement. Lumbar extension improved immediately and in the short term in both the groups; however, lumbar flexion showed improvement only in the SNAGs group over the short-term but not immediately. Lateral flexion ROM did not show any significant change for both the groups (Table 4).

Table 4 Time*group analysis for VAS, PSFS and ROM for MFR and SNAGs groups.

Variables	Group	Factors	Mean Difference	Std. Error	p-value	95% Confidence Interval	
VAS(mm)	MFR	Pre * Immediate	2.503	0.344	0.0001**	1.63	3.37	
Pre * Post	3.806	0.309	0.0001**	3.02	4.58	
SNAGs	Pre * Immediate	2.484	0.0322	0.0001**	1.66	3.3	
Pre * Post	3.544	0.315	0.0001**	2.74	4.34	
PSFS	MFR	Pre * Immediate	−1	0.175	0.0001**	−1.441	−0.559	
Pre * Post	−2.156	0.197	0.0001**	−2.655	−1.658	
SNAGs	Pre * Immediate	−1.495	0.212	0.0001**	−2.031	−0.959	
Pre * Post	−2.751	0.278	0.0001**	−3.454	−2.047	
Flexion (degrees)	MFR	Pre * Immediate	−0.545	1.112	1	−3.35	2.26	
Pre * Post	−1.879	0.894	0.13	−4.13	0.379	
SNAGs	Pre * Immediate	−1.875	1.24	0.422	−5.01	1.26	
Pre * Post	−4.594	1.489	0.013**	−8.36	−0.82	
Extension(degrees)	MFR	Pre * Immediate	−6.03	1.234	0.0001**	−9.14	−2.91	
Pre * Post	−7.455	1.258	0.0001**	−10.63	−4.27	
SNAGs	Pre * Immediate	−5.03	1.522	0.007**	−8.88	−1.17	
Pre * Post	−7.219	1.42	0.0001**	−10.81	−3.62	
Left lateral flexion (degrees)	MFR	Pre * Immediate	−1.27	1.366	1	−4.72	2.17	
Pre * Post	−0.97	1.413	1	−4.54	2.6	
SNAGs	Pre * Immediate	−0.375	0.912	1	−2.68	1.93	
Pre * Post	−1.688	1.283	0.594	−4.93	1.56	
Right lateral flexion (degrees)	MFR	Pre * Immediate	−2.242	1.059	0.126	−4.91	0.43	
Pre * Post	−2.758	1.416	0.181	−6.33	0.82	
SNAGs	Pre * Immediate	−1.06	1.033	0.934	−3.67	1.55	
Pre * Post	−1.594	1.19	0.57	−4.6	1.41	
Notes.

** p < 0.05 statistical significant.

VAS Visual Analog Scale

mm millimetre

MFR Myofascial release

SNAGs Sustained Natural Apophyseal Glides

PSFS Patient Specific Functional Scale

Comparison between the groups for VAS, PSFS, and ROM showed no statistically significant difference (p > 0.05) immediately and also in the short-term, including MODI (Table 5).

Table 5 Between-group analysis of VAS, MODI, PSFS, and ROM: immediate and over short-term.

Variable	MEDIAN (IQR)	p-value	
	MFR	SNAGs		
Immediate	
VAS (mm)	4.1(1.9–5.2)	3.05(2.05–4.37)	0.328	
Flexion (degrees)	50 (45–59)	52.5 (45–59.5)	0.937	
Extension(degrees)	20 (15–30)	21.5 (18.25–29.5)	0.889	
Left lateral flexion (degrees)	22 (15–30)	20 (15–25)	0.595	
Right lateral flexion (degrees)	20 (17.5–26)	20 (15–27.75)	0.889	
PSFS	5.66(4.49–6.16)	5.33(5–6.66)	0.388	
Short-term	
VAS (mm)	2(1.0–3.45)	2(0.85–3.65)	0.674	
Flexion (degrees)	52 (45.5–60)	55.5 (50–60)	0.380	
Extension(degrees)	25 (19–30)	25 (20–31.5)	0.731	
Left lateral flexion (degrees)	20 (20–25)	24.5 (15–28)	0.754	
Right lateral flexion (degrees)	20 (17.5–26)	20 (15.75–28)	0.931	
MODI (%)	12(6–16)	8(6–15.5)	0.472	
PSFS	7(5.83–7.58	7(6.12–7.62)	0.385	
Notes.

* p < 0.05 statistical significant.

Discussion

This study was aimed to determine the comparative effects of MFR and SNAGs in combination with strengthening exercises on pain, disability, ROM, and functional ability in NSLBP patients. However, both MFR and SNAGs groups demonstrated statistically significant (p < 0.05) improvements for outcomes VAS, ROM, and PSFS, immediately and in the short-term, including MODI, there were no significant differences between the groups (p > 0.05).

The clinically significant improvement was observed on VAS for pain in both SNAGs and MFR group with strengthening exercises. Considering the MCID value of 20 mm for VAS in chronic pain (Vela, Haladay & Denegar, 2011), SNAGs groups demonstrated a significant change of 30.5 mm immediately after the 1st treatment session and 41 mm over the short-term. In comparison, an MFR group improved by 21 mm immediately and 42 mm in the short-term.

Similarly to this study’s findings, when Waqqar et al. compared SNAGs with other manual therapy techniques like McKenzie exercises, they found both have a similar effect for pain and disability (Waqqar, Shakil-Ur-Rehman & Ahmad, 2016). Other studies also found SNAGs (Tul Ain et al., 2019) or MFR (Ajimsha, Daniel & Chithra, 2014; Tozzi, Bongiorno & Vitturini, 2011; Saratchandran, 2013; Arguisuelas et al., 2017; Arguisuelas et al., 2019), when administered as an adjunct to occupational therapy, stretching, back strengthening exercises, and thoracic postural correction exercises, have short-term beneficial effects on NSLBP. This study’s findings indicate that both techniques with exercise can be used to address NSLBP. Though both groups improved similarly over the short-term in the present study, SNAGs were clinically superior to improve pain immediately. A study by Rezkhallah et al. has reported similar findings in non-specific neck pain patients. In their research, SNAGs improved pain with more percentage points than MFR (Rezkallah & Abdullah, 2018).

Both MFR and SNAGs help in correcting anomalies within the elements of the movement system by stimulating mechanoreceptors, which might improve the activation pattern of para-spinal muscles, improving pain-free ROM. However, in the current study, only the SNAGs group improved with lumbar flexion ROM, while lumbar extension improved for both the groups. This finding contradicted the result in non-specific neck pain patients in which both MFR and SNAGs improved neck ROM in all the planes (Rezkallah & Abdullah, 2018).

In line with the finding of this study, SNAGs application had an immediate and short term effect on lumbar flexion ROM among healthy individuals (Moutzouri et al., 2008) and patients with NSLBP (Waqqar, Shakil-Ur-Rehman & Ahmad, 2016; Khan, Torairi & Shamsi, 2018; Elrazik et al., 2016; Tul Ain et al., 2019; Muhanna, 2018). Passively administered spinal accessory glide breaks adhesions, leading to increased facet joint vascular supply and necessary nutrients, enhancing the soft tissue healing around the injury site (Rezkallah & Abdullah, 2018). The application of the glides over the spinous process concentrated on correcting flexion and extension positional faults and promoting pain-free physiological lumbar spine movement (Pourahmadi et al., 2018). In this study, Mulligan mobilization was delivered with a contact of the spinous process, which glides both facets in the same direction. During mobilization, patients also performed only sagittal plane movements of lumbar flexion or extension, as it was primarily restricted movement. We hypothesize, this reason for no observed improvement in lateral flexion, as it requires ipsilateral facet to move in extension with contralateral facet moving to flexion and can be better-improved giving mobilization using unilateral transverse process.

MFR found to normalize flexion relaxation phenomenon in individuals with NSLBP (Arguisuelas et al., 2019), which contradicted the observation of this study in which the MFR group did not improve with lumbar flexion ROM. However, other studies have reported that MFR as an adjunct to back school along with exercises (Saratchandran, 2013) and work station modifications (Balasubramaniam, Ghandi & Sambandamoorthy, 2013) improved lumbar flexion ROM. These outcomes could be related to more MFR sessions, which helps break down the scar matrix and intermolecular crosslinks, redistribute internal fluids, and improve collagen extensibility. These effects of MFR may help in enhancing fascial mobility and soft-tissue extensibility (Balasubramaniam, Ghandi & Sambandamoorthy, 2013).

In this study, MODI demonstrated statistical (p < 0.05) significant difference for both MFR and SNAGs groups from baseline to short-term. However, only the SNAGs group shown clinically significant improvement. The MCID value for the MODI has been estimated to be six points (Fritz & Irrgang, 2001). Over the short-term, the MFR group showed a gain of four points, while a change of six points was observed in the SNAGs group. Rezkhallah et al. reported similar findings with the SNAGs group improved more than the MFR group in non-specific neck pain patients. The more significant gain observed in pain and ROM can explain the improvement in disability amongst the SNAGs group compared to the MFR group.

Both the groups demonstrated improved function immediately and over short-term, but between the group, there was no significant difference for PSFS. The change for the PSFS was 2.67 in both MFR and SNAGs groups over the short-term, which was not clinically significant. Abbott (2014) and Van Vliet, Hefford & Abbott (2012) reported MCID values of 3.3 and 4.3, respectively, for more considerable clinical change of PSFS in chronic mechanical LBP.

Manual therapy interventions like MFR and SNAGs stimulate mechanoreceptors located in the soft tissues and the lumbar spine’s facet joints. The activity of these receptors constantly feeds CNS for neuro-reflexive muscle activation. A possible explanation that both MFR and SNAGs groups showed similar improvement could be related to these techniques’ effect on seducing CNS by balancing these receptors’ activity and re-establish dynamic control.

Limitation

Severe limitation of this trial was the non-random allocation of patients to treatment arms due to the trial’s lack of insurance cover.

The study could not complete the estimated sample size recruitment within the data collection time-frame.

Conclusions

The current study results suggest that strengthening exercises with SNAGs or MFR techniques can be considered for immediate and short-term management of pain, function, and lumbar extension ROM in sub-acute to chronic NSLBP. Future trials should consider assessing the long-term effects of SNAGs and MFR for improvement in lumbar ROM. Varying duration of MFR hold also needs to be studied with a long-term follow-up.

Supplemental Information

Supplemental Information 1 Raw data

Click here for additional data file.

Supplemental Information 2 CONSORT checklist

Click here for additional data file.

Supplemental Information 3 Trail protocol

Click here for additional data file.

Additional Information and Declarations

Competing Interests

Author Contributions

Human Ethics

Clinical Trial Ethics

Data Availability

Clinical Trial Registration

Manisha P. Shenoy is employed as a physiotherapy specialist at Hamad Medical Corporation, Doha, Qatar.

Vignesh Bhat P and Vivek Dineshbhai Patel conceived and designed the experiments, performed the experiments, analyzed the data, prepared figures and/or tables, authored or reviewed drafts of the paper, and approved the final draft.

Charu Eapen and Steve Milanese analyzed the data, prepared figures and/or tables, authored or reviewed drafts of the paper, and approved the final draft.

Manisha Shenoy conceived and designed the experiments, authored or reviewed drafts of the paper, and approved the final draft.

The following information was supplied relating to ethical approvals (i.e., approving body and any reference numbers):

Ethical approval for this study was obtained from the Institutional Ethics Committee, Kasturba Medical College, Mangalore to conduct the study in its two tertiary care medical college hospitals (iec kmc mlr 11-18/429).

The following information was supplied relating to ethical approvals (i.e., approving body and any reference numbers):

Ethical approval for study was obtained from Institutional Ethics Committee, Kasturba Medical College, Mangalore to conduct study in its two tertiary care medical college hospitals.

The following information was supplied regarding data availability:

Raw data, including demographic details and outcome measure values, are available in the Supplemental Files.

The following information was supplied regarding Clinical Trial registration:

CTRI/2018/12/016787.

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
