# Peer review of "Myofascial release versus Mulligan sustained natural apophyseal glides’ immediate and short-term effects on pain, function, and mobility in non-specific low back pain"

_PeerJ, doi:10.7717/peerj.10706_

## Round 0.1 · original submission · Major Revisions

I have received three reviews of your work. They all find that it could be an exciting contribution. However, a lot of work must be done to make the manuscript publishable. As you will see, the reviewers have done a detailed job. I am particularly concerned about the objections made to your methodological strategy, which lacks sufficient detail. According to our reviewer, you should report the evidence behind the effectiveness of both SNAGs and MR using recent systematic reviews. The reviewers also asked several questions and identify flaws in the paper that you need to take in full consideration, including even the language issue.
Please pay attention to the annotated pdfs. Thank you.

·

Basic reporting

first of all, i would like to thank the authors for their efforts and interesting manuscript.
yet i have few general comment:
1- the English language used was acceptable, and clear, yet it need extra revision by a native English person.
2- the literature was sufficient and cover adequate knowledge about the topic.
3- figures presented are not in a good quality regarding the resolution. and need some modifications as i recommended in the attached annotated pdf file.
4- the tables missed some data, and i posted some comments on this issue.
5- the article structure is good, and my single comment here is about the lack of randomization. however, i posted few comment i believe they will increase the strengths of the article.

Experimental design

no comment

Validity of the findings

no comment

·

Basic reporting

No

Experimental design

No

Validity of the findings

No

Additional comments

The findings of this study add on support to the existing literature. Future research could concentrate on different perspectives of the treatment and outcome measures.

Reviewer 3 ·

Basic reporting

the articles fails to meet your standards in introduction and discussion. too long paragraphs as it were not concise and clear.

Experimental design

no comment

Validity of the findings

no comment

Additional comments

your work is fine but needs rewriting to get into point and to justify your study without writing many mechanisms to explain your idea

Annotated reviews are not available for download in order to protect the identity of reviewers who chose to remain anonymous.

---

## Round 0.2 · Minor Revisions

I apologize for my delay in make my decision. Some details remain before the manuscript can be accepted for publication. Please take into consideration the last reviewers' suggestions. In line 86 of the Introduction section you wrote "restricted ROM", However, you defined ROM as restricted range of motion, please check this point.

·

Basic reporting

good

Experimental design

good

Validity of the findings

good

Additional comments

i would like to thank the authors for their patience and work fro improving their manuscript.
my single comment here is regarding the footnotes of the tables, its still not written correctly. foot notes used to explain any abbreviation mentioned inside the table even if it was explained in the main text (the table is an idependant identity). so please add the explanation (full name ) for every letter or abbreviation in the table
examples:
SNAGs, sustained natural .....
VAS, visual analog scale
n, number

---

## Round 0.3 · accepted · Accept

Thank you for your consideration of our last suggestions.